# Cognitive Fatigue in Multiple Sclerosis: An Objective Approach to Diagnosis and Treatment by Transcranial Electrical Stimulation

**DOI:** 10.3390/brainsci9050100

**Published:** 2019-05-02

**Authors:** Stefanie Linnhoff, Marina Fiene, Hans-Jochen Heinze, Tino Zaehle

**Affiliations:** 1Department of Neurology, Otto-von-Guericke University Magdeburg, Leipziger Street 44, Magdeburg 39120, Germany; stefanie.linnhoff@med.ovgu.de (S.L.); heinze@med.ovgu.de (H.-J.H.); 2Department of Neurophysiology and Pathophysiology, University Medical Center Hamburg-Eppendorf, Martinistraße 52, Hamburg 20246, Germany; m.fiene@uke.de; 3Center for Behavioral Brain Sciences, Magdeburg 39106, Germany

**Keywords:** cognitive fatigue, multiple sclerosis, objective measurement, fatigability, tDCS, tACS

## Abstract

Cognitive fatigue is one of the most frequent symptoms in multiple sclerosis (MS), associated with significant impairment in daily functioning and quality of life. Despite its clinical significance, progress in understanding and treating fatigue is still limited. This limitation is already caused by an inconsistent and heterogeneous terminology and assessment of fatigue. In this review, we integrate previous literature on fatigue and propose a unified schema aiming to clarify the fatigue taxonomy. With special focus on cognitive fatigue, we survey the significance of objective behavioral and electrophysiological fatigue parameters and discuss the controversial literature on the relationship between subjective and objective fatigue assessment. As MS-related cognitive fatigue drastically affects quality of life, the development of efficient therapeutic approaches for overcoming cognitive fatigue is of high clinical relevance. In this regard, the reliable and valid assessment of the individual fatigue level by objective parameters is essential for systematic treatment evaluation and optimization. Transcranial electrical stimulation (tES) may offer a unique opportunity to manipulate maladaptive neural activity underlying MS fatigue. Therefore, we discuss evidence for the therapeutic potential of tES on cognitive fatigue in people with MS.

## 1. Introduction

Multiple sclerosis (MS) is a chronic inflammatory disease of the central nervous system that leads to demyelination and atrophy of brain cells and has a profound impact on motor functioning and cognition. Worldwide the median prevalence is 33 per 100.000 people suffering from MS, with women being twice as often affected than men [1]. MS is a very diverse disease with heterogeneous clinical symptoms. Depending on the area of inflammation and resulting lesions, various phenotypically different neurological deficits may occur. 

Among frequently reported deficits, fatigue remains one of the most common and challenging symptoms in MS affecting up to 75% of patients [2,3]. The syndrome includes a lack of motivation, an overall feeling of exhaustion as well as behavioral performance decrements, and is the main reason for early retirement in people with MS [4]. The exact pathogenic mechanisms underlying MS fatigue are yet not fully understood. Particularly three influential core hypotheses have been proposed. Accordingly, fatigue has been related to (1) neuroimmune dysregulation based on increased levels of inflammatory mediators such as interferon or interleukin, (2) neuroendocrine dysfunction resulting in hyperactivation of the hypothalamo–pituitary–adrenal axis, and (3) demyelination, cortical lesions, and functional brain abnormalities within various cortical and subcortical brain regions (see Ayache & Chalah [5] for a review of studies on the pathogenesis of MS fatigue). The latter hypothesis is supported by a large number of neuroimaging studies proposing a malfunctioning cortico–striato–thalamo–cortical network, the so called fatigue circuit, underlying MS fatigue [6,7]. Hence, various previous research demonstrated relations between subjective trait-fatigue and structural and functional abnormalities in the frontal regions [8,9,10,11,12], parietal regions [8,13,14], corpus callosum [15,16,17], basal ganglia [10,18,19,20], and thalamus [18,19].

Regarding its diagnosis, the fatigue construct has been divided into a motoric, psychosocial, and cognitive dimension [2]. In this review, we will focus specifically on the assessment and therapy of the latter dimension. Cognitive fatigue significantly impairs daily life and is just as debilitating to people with MS as motoric fatigue. However, the concept of cognitive fatigue is still only poorly understood. According to the multidimensional nature of MS fatigue, various definitions exist in the current literature. The MS Council [21] defines MS fatigue in general as a “subjective lack of physical and/or mental energy that is perceived by the individual or caregiver to interfere with usual and desired activities” [21] (p. 2), which specifically emphasizes the current subjective understanding of the syndrome. As a result, multiple self-report questionnaires assessing the severity of fatigue, such as the Fatigue Severity Scale (FSS) [22], the Fatigue Impact Scale (FIS) [2], the Modified Fatigue Impact Scale (MFIS) [21], the Fatigue Scale for Motor and Cognitive Functions (FSMC) [23], or the Wuerzburg Fatigue Inventory for Multiple Sclerosis (WEIMuS) [24] have been developed. While FSS is only a one-dimensional questionnaire, the other four evaluate distinct fatigue dimensions, including cognitive fatigue. Importantly, although these questionnaires are extensively used to diagnose cognitive fatigue, they exclusively assess the subjective experience of people with MS. Yet, subjectively assessed parameters are retrospective statements and therefore mood-sensitive and subject to psychological errors, such as regression to the mean or recall bias which reduce their diagnostic accuracy [25]. Additionally, these questionnaires show low correlations among each other and heterogeneous associations to patients’ functional impairment, disease duration, or cognitive deficits [14,18,26,27]. Thus, for comprehensive clinical diagnostics of cognitive fatigue, assessment of subjective exhaustion needs to be complemented by the objectively measurable impact of fatigue on patients’ functioning. As suggested by Holtzer et al. [28], this objective cognitive fatigue can be assessed as behavioral consequences of “an executive failure to maintain and optimize performance over acute but sustained cognitive effort” as this will result “in performance that is lower and more variable than the individual’s optimal ability” [28] (p. 108). Hence, according to its definition cognitive fatigue must be operationalized as strong performance decrements in cognitive demanding tasks over time, rather than as current performance at only one measurement time point, as the latter might only reflect the level of overall cognitive impairment.

The utilization of a unified taxonomy and its precise use in research communication is of particular importance for future progress in MS-related fatigue research. In Figure 1, we propose a generally valid fatigue taxonomy. Summarizing former suggestions, fatigue can be subdivided into physical, psychosocial, and cognitive fatigue [2]. While psychosocial fatigue is only subjectively measurable, physical and cognitive fatigue can be assessed subjectively as well as objectively. Specifically, subjective cognitive fatigue refers to an ongoing perceived feeling of exhaustion. Objective cognitive fatigue—hereafter referred to as fatigability—describes a performance decline during cognitive tasks, measurable through the change in cognitive performance relative to a baseline [29]. Subjective and objective cognitive fatigue can be further subdivided. Subjective fatigue divides into a trait and a state component. Trait-fatigue refers to a global status of the patient that changes slowly over time, while state-fatigue means the change in subjectively perceived fatigue level over time [30]. Accordingly, subjective trait-fatigue can be evaluated through self-questionnaires and subjective state-fatigue through visual analogue scales (VAS) or numerical rating scales. In contrast, objective fatigue (fatigability) is per definition state-dependent and enables an objective assessment by behavioral or electrophysiological parameters that will be explained in detail in the following section. Thus, the proposed concept of cognitive fatigue implies that it can be studied both qualitatively as a subjective phenomenon and quantitatively as an objective phenomenon.

## 2. Search Strategies 

In order to give an exhaustive overview of the literature, we searched for relevant studies in English and German languages addressing MS fatigue on electronic databases (i.e., PubMed, Scopus, and Cochrane database), until the end of January 2019. The following research terms and cross-combinations of the terms were used: “multiple sclerosis” or “MS”, “fatigue”, “fatigability”, “cognitive fatigue”, “objective fatigue”, “performance decrement”, “time on task”, “noninvasive brain stimulation”, and “transcranial direct current stimulation” or “tDCS” and “transcranial alternating current stimulation” or “tACS”. Further, we also scanned the references of the selected studies in order to look for additional relevant sources. 

## 3. Objective Measurement of Cognitive Fatigue

To overcome the purely subjective character of fatigue diagnostics, recent research focused on the investigation of objective diagnostic measures of fatigability in people with MS. The methods to operationalize fatigability as performance reduction with time-on-task can be divided into four approaches [31]. The first approach is to investigate fatigability over a prolonged period of time, in which the subjects perform the test paradigm several times in a row and performance changes are compared to a baseline. Applying this approach, some studies reported evidence for a fatigue-related performance decline [25,32,33], while others did not [34,35,36,37]. The second and third approach define fatigability as a pre-to-post performance decline in a specific task A while inducing fatigue by mental (second approach) or physical exertion (third approach) in a task B in between. However, evidence in support of these approaches is rare and inconsistent [38,39]. The fourth and most promising method is to measure fatigability during sustained mental effort and to compare the performance at the beginning of a cognitively demanding task with performance level at the end. Using this approach, fatigability has been repeatedly demonstrated [40,41,42,43,44,45,46]. 

Fatigue has been shown to become most prominent during sustained attention tasks that depend on a high level of endogenous attention. Accordingly, subjective fatigue shows strong relations to performance decline in alertness and vigilance tasks [25,33,39,40,43,45,47,48]. In contrast, it does not impair memory performance, language, or visuospatial processing [34,36,38,49] and shows only weak associations with performance decline in tasks on processing speed [34,35,37,41,44,50,51] and working memory [32,52,53]. According to Hanken et al. [54], only alertness or vigilance tasks require maintained intrinsic attention over a prolonged period of time that can easily be distracted by interoceptive events or mind wandering, which can result in cognitive fatigability. This relation is further supported by an overlap of neural alterations in the fatigue circuit and brain regions involved in attentional processing [16,55,56,57,58].

In the following, we will present a series of objective parameters that have been proven to be suitable surrogate markers for assessing fatigability. Table 1 presents an overview of studies investigating objective cognitive fatigue in people with MS, sorted by the used approaches to measure fatigability. 

### 3.1. Behavioral Measures

Behaviorally, fatigability can be assessed through changes in reaction time, accuracy, and processing speed in simple alertness or vigilance tests over time. There are numerous studies showing increasing reaction times [25,33,38,39,45,47,59,60] and decreasing accuracy [32,41,42,43] with time-on-task, mostly assessed by administering simple reaction time tests such as the alertness subtest of the Test Battery for Attentional Performance (TAP) [61]. Claros-Salinas et al. [33] measured fatigue in the TAP alertness task at three different time points during the course of one day. While subjective state-fatigue increased diurnally in both, participants with MS and healthy controls, cognitive performance decreased over the day only in the MS group. Furthermore, performance changes in the TAP alertness task were assessed at baseline and after 2.5 hours of physical and cognitive exertion. While healthy controls improved from first to second test administration, performance of the MS group decreased [47]. Similarly, Neumann et al. [39] investigated fatigability by measuring the alertness level before and after participants performed a cognitively demanding task. The authors found increased reaction times after cognitive load only in people with MS, while reaction time remained unchanged in healthy controls. Thus, these studies indicate that objective cognitive fatigue parameters are well suited for assessing MS-related fatigue pathology. 

It is noteworthy, however, that there are also studies showing no susceptibility of reaction time [32,52,62] or accuracy performance [63,64] to cognitive fatigue. Therefore, some authors suggested finer-grained analytical methods like reaction time variability, which is defined as the standard deviation of correct response times or the coefficient of variation, which is calculated by dividing the standard deviation by the mean reaction time and thus avoiding confounding effects of group differences in mean reaction times [36,65]. Cognitive fatigue may lead to occasional lapses in attention followed by higher reaction time variability even in the absence of a linear time-on-task decline [36]. Accordingly, analyses accounting for individual variability might be more sensitive in diagnosing behavioral fatigability effects. 

Besides reaction times, cognitive processing speed and working memory changes can further be indicators for cognitive fatigue declines. They are commonly assessed using the Paced Auditory Serial Addition Test (PASAT) [66] or the Signal Digit Modalities Test (SDMT). Studies utilizing the PASAT to measure fatigability typically report a significant performance reduction from the first through the second half of the task [41,44,50,53]. A more fine-grained analysis is the percent dyad score method suggested by Snyder et al. [67]. This score only counts the total number of two correct responses in a row proposed as a better estimate of performance correctness according to the intended task demands. In line with this assumption, one study showed that the total number of correct responses in the PASAT did not differ between participants with MS and healthy controls, while when examining percent dyad score, only the MS group showed pronounced susceptibility to cognitive fatigability [41]. 

### 3.2. Electrophysiological Measures

Recording of brain activation by electroencephalography (EEG) and event-related potentials (ERP) has been proven as a sensitive method for the objective assessment of neural alterations related to cognitive fatigue. Specifically, the P300 ERP is widely used as an index of cognitive functioning [73,74]. The component is generally evoked in an oddball paradigm, when rare target stimuli are presented in a sequence of standard stimuli. The P300 amplitude is proposed to be proportional to the amount of attentional resources devoted to a given task, while P300 latency indicates processing speed [75]. 

Previous studies demonstrated longer latencies and smaller amplitudes of P300 component in people with MS [76,77]. Pokryszko-Dragan et al. [78] investigated changes in P300 and cognitive performance in patients and found prolonged latencies and reduced amplitude of P300 associated with increased subjective cognitive fatigue. Chinnadurai et al. [42] conceptualized fatigability as the ratio between the processing of the first and last items in an ongoing oddball paradigm and evaluated P300 alterations. As a result, participants with MS showed prolonged P300 latencies for the last 50 items compared to first 50 items. Regarding P300 amplitude, data revealed no significant difference between people with MS and healthy controls. In our recent interventional study [25], patients with subjective fatigue performed three blocks of an auditory oddball paradigm to assess cognitive fatigability. The MS group that did not receive an intervention showed fatigability-related increased P300 latencies and decreased amplitudes with time-on-task.

### 3.3. Sensory Gating Parameter

Sensory gating plays a key role in cognitive control and attention. It protects stimulus processing from interference caused by subsequent incoming information. Sensorimotor gating can be measured using prepulse inhibition (PPI). PPI means a reduced startle response to an intense stimulus, when a low intensity stimulus (prepulse) is presented beforehand. Van der Linden et al. [79] investigated 20 healthy subjects that were randomly allocated to a fatigue or non-fatigue condition. Before and after a cognitively demanding task, PPI was evaluated. Results showed a significant reduction in PPI during cognitive fatigue state. Thus, induction of cognitive fatigue by a cognitively demanding task negatively affected sensorimotor gating. Additionally, the reduction in PPI correlated positively with subjective state-fatigue evaluated by VAS. Another sensory gating parameter is the event-related potential P50. The P50 is generally evoked using the auditory paired click paradigm, when one click sound is followed by a second click sound approximately 500 ms after the first one. The processing of the first stimulus suppresses processing of the second stimulus, thereby leading to decreased P50 amplitude to the second click. One study by Aleksandrov et al. [80] examined the P50 before and after inducing cognitive fatigue by muscle load. Data showed that physical exertion significantly decreased or completely suppressed the sensory gating index. However, no study systematically investigated PPI and P50 changes through fatigability in people with MS so far. Whether the diagnostic value of sensory gating parameters shown in clinically non-significant fatigue in healthy subjects can be generalized to pathological MS fatigue needs to be further investigated. 

Taken together, fatigability is best operationalized with sustained attention tasks measuring alertness or vigilance declines over time. Sustained attention tasks like the TAP alertness subtest, PASAT, and SDMT have been proven to reliably lead to objectively measurable performance declines in people with MS. Other cognitive domains like memory, language, or visuospatial processing, verbal learning or working memory do not seem to be consistently affected by cognitive fatigue. Parameters that have been shown to represent objective indices for fatigability in people with MS are simple reaction time and accuracy, as well as the P300 ERP. Fatigability consistently leads to increasing reaction times, decreasing accuracies, and smaller amplitudes as well as longer latencies in the P300 ERP component. Additionally, recent studies present finer-grained analyses (i.e., response time variability, coefficient of variant, or percent dyad score), as more sensitive measures of performance decline over time. To reliably measure clinically significant fatigability in people with MS, it is however important to differentiate between patients with and without fatigue. Fatigue-related strong performance deteriorations can otherwise not be distinguished from typical performance decrements over time that might also occur in healthy subjects. Common approaches to determine the clinical significance of fatigue symptoms are the definition of cut-off values on subjective fatigue questionnaires or the investigation of statistically significant differences in fatigability between patients and healthy controls [29]. Moreover, closer investigations on the relation between the level of objective cognitive fatigue and demographic characteristics of patients (e.g., disease duration or disability status), can help to understand the implications of this symptom for patients’ daily functioning during the course of the disease. These aspects should be considered in future studies on cognitive fatigue. 

## 4. Relationship between Objective and Subjective Fatigue

The relation between subjective fatigue and fatigability is still a topic of controversy. If cognitive fatigue affects task performance of people with MS, it should be paralleled by subjectively perceived fatigue. However, there are numerous studies showing no relationship between subjective and objective cognitive fatigue measures [32,35,41,68,81,82,83,84,85,86,87]. One reason for this observed divergence might be related to the high heterogeneity in the diagnostic scales used for assessing subjective cognitive fatigue. Additionally, in most studies subjective fatigue questionnaires mainly measure the trait component of fatigue by assessing the impact of fatigue on daily activities over the past weeks. However, since fatigability is defined as a performance decrement over time, it might be more related to changes in subjective state-fatigue during the course of a testing session. In the following, we discuss this aspect in more detail by differentiating between studies measuring subjective trait- or state-fatigue and correlating these values with overall mean performance or changes in performance. 

Of those studies investigating the relationship between subjective trait-fatigue and mean cognitive performance during a fatiguing task, three found a positive correlation between FSS score and reaction time [34,88] or P300 ERP [78], and another three studies reported a positive relation specifically between cognitive fatigue and reaction times [36,39,89]. Two studies found no associations [41,84]. Five studies examined the relationship between subjective trait-fatigue and fatigability indicated by performance decrement over time. While one study reported no association [44], three studies found a positive relationship [40,47,53], indicating slower processing speed and longer reaction times as the task progressed with greater subjective trait-fatigue. Schwid et al. [50] found a significant correlation between the FSS questionnaire and performance change from the first to second half in the PASAT, but no correlations between performance decline and the MFIS cognitive fatigue subscale. Five studies especially examined subjective state-fatigue and their relationship to objective performance decline with time-on-task. Four studies found a positive relationship [25,43,47,60], while only one study did not [32]. Hence, longer reaction times with time-on-task as well as more omissions in the second half were shown to be associated with a greater feeling of momentary exhaustion [43,47,60]. Finally, in a recent study we further revealed a positive association between subjective state-fatigue assessed by a VAS and P300 latency and a negative relation with P300 amplitude [25].

Due to this heterogeneity in the literature, the relationship between subjective and objective fatigue measures still remains unclear. Assuming that in some patients subjective and objective fatigue jointly appear, correlations might only be detectable when choosing valid fatigue parameters. For objective fatigue measures, this might include the change in simple reaction time, accuracy, and ERPs. However, not only the choice of objective fatigue markers, but also the reliable assessment of subjective fatigue changes over time is a challenging methodological factor. Likert rating scales are frequently used, but have limited variability that hampers the detection of correlations with objective fatigability measures. Based on former studies, Hanken et al. [90] proposed a theory inclining the subjective feeling of fatigue and the objectively measurable fatigability into one model. They proposed that subjective fatigue results from inflammation-induced sickness behavior and altered neural processing within interoceptive and homeostatic brain areas, including the insula, the anterior cingulate, and the hypothalamus. Via increased interoceptive interference, subjective fatigue might secondarily lead to objective fatigue symptoms in terms of measurable performance decrements. Importantly, the latter can even be exaggerated by cortical atrophy in the alerting system, thereby accounting for the relevance of the attention network contributing to the pathogenesis of objective cognitive fatigue. Hence, according to their model, objectively measurable performance changes can also exist independent of subjective fatigue due to cortical atrophy in the attention network, which might explain variability in correlations between subjective and objective fatigue in the current literature. These considerations demonstrate the importance of including both subjective and objective fatigue in a holistic fatigue concept and emphasize the use of a clear and unified taxonomy in future fatigue research.

## 5. Therapeutic Potential of tES for Cognitive Fatigue

As MS-related cognitive fatigue drastically affects a patient´s quality of life, the development of efficient therapeutic methods for overcoming fatigue is of high clinical relevance. Especially for a systematic treatment evaluation and optimization, a reliable and valid assessment of the individual fatigue level by objective parameters is essential. Transcranial electrical stimulation (tES) may offer a unique opportunity to manipulate the maladaptive neural activity underlying MS fatigue. The neuromodulatory potential of tES is widely shown on cognitive, perceptual, and motor processes [91]. As changes in brain activity were demonstrated in various neurological and psychiatric conditions, the clinical application of tES has been increasingly progressed with the aim to restore pathological brain function and to improve related symptoms [92]. In the following, we will first discuss evidence for the therapeutic potential of transcranial direct current stimulation (tDCS) on cognitive fatigue in people with MS. Moreover, we will emphasize the functional importance of altered neural oscillatory pattern in fatigue pathogenesis and discuss the possible advantage of transcranial alternating current stimulation (tACS) application for cognitive fatigue treatment. Table 2 presents an overview of studies evaluating tES effects on objective cognitive fatigability in people with MS and healthy controls. 

### 5.1. Neuromodulation of the Fatigue Circuit by tDCS

tDCS is one of the most frequently used tES techniques that delivers a constant, low-intensity electrical current to the brain, resulting in modulation of cortical excitability [100]. The current is steadily flowing between two or more surface electrodes (anode and cathode) placed on the scalp. The external electric field forces a shifting in intracellular ions in cortical pyramidal cells, thereby modifying internal charge and resting membrane potential. The stimulation-induced effects of current flow parallel to the somatodendritic axis in the target region depend on current polarity. Generally, anodal tDCS enhances cortical excitability via depolarization of resting membrane potentials, whereas cathodal tDCS decreases cortical reactivity via hyperpolarization of neuronal membranes [101]. Excitability-enhancing effects of anodal tDCS have been successfully demonstrated to outlast the stimulation period by several minutes to hours proposed to result from long-term synaptic changes in the stimulated region [102,103].

In healthy participants, tDCS over the left dorsolateral prefrontal cortex (DLPFC) has been shown to mitigate fatigue-induced decrements in vigilance performance over time [94,95,96]. McIntire et al. [95] showed that tDCS was more beneficial than caffeine consumption in counteracting subjective state-fatigue and objective vigilance task decline during prolonged wakefulness. While these studies suggested tDCS as an effective fatigue countermeasure to maintain vigilance performance, one study failed to show effects of frontal tDCS on performance decline in a high cognitively demanding working memory task over time [93]. Beside stimulation of prefrontal brain regions, bilateral tDCS over the parietal cortex has further been shown to prevent fatigability in visual detection performance in healthy subjects [97]. For pathological MS-related fatigue, several studies have investigated the efficacy of tDCS over the fatigue circuit with the aim to restore altered neural excitability and improve subjective exhaustion. Positive effects of anodal tDCS over the left DLPFC [104,105,106,107], the bilateral primary somatosensory cortex [108,109,110], or the bilateral primary motor cortex [111] were shown on subjective trait- and state-fatigue in people with MS. These studies gave important insight into the causal relevance of the targeted brain regions in fatigue pathogenesis. However, tDCS-induced improvements in MS-related objective cognitive fatigue parameters have rarely been the focus of investigation. Recently, Hanken et al. [48] examined tDCS effects on fatigability measured as vigilance decrements with time-on-task in MS. Results demonstrated that anodal stimulation over the right parietal cortex as part of the vigilance network delivered for 20 min could counteract the reaction time increase during prolonged testing compared to sham. Yet, subjective state-fatigue increased independent of stimulation condition. Likewise, we investigated effects of tDCS on cognitive fatigue-associated behavioral and electrophysiological parameters in people with MS [25]. We showed that anodal tDCS of the left DLPFC for about 30 min caused an increase in P300 amplitude that persisted after the end of stimulation and reduced the fatigability-related increase in reaction time over the course of the testing session in comparison to sham. However, in line with the study by Hanken et al. [48], stimulation did not counteract the increase in subjective state-fatigue with time-on-task. This dissociation between the feeling and the behavioral characteristics of fatigue might suggest that while a single session of anodal tDCS could lead to improvements in objective fatigability parameters, multiple repetitive sessions might be necessary to induce cumulative changes in the fatigue network that lead to subjectively perceivable changes in the feeling of fatigue [106,107,108,109,110,111,112,113]. Therefore, stimulation dosage and duration are presumably critical aspects that need to be considered for the development of effective stimulation protocols targeting subjective and objective fatigue symptoms.

### 5.2. Role of Neural Oscillations in Cognitive Fatigue and its Modulation by tACS

Cognitive fatigue has not only been associated with altered neural excitability, previously targeted by tDCS, but has also been related to alterations in neural oscillatory activity [114,115]. In healthy subjects, a systematic shift from fast to low frequency waves has been reported during a reduced level of arousal [116]. Cognitive fatigability in healthy subjects has been repeatedly shown to be associated with power increase in the theta (4–8 Hz) and alpha (8–14 Hz) frequency band over frontal, central, and parietal brain regions with time spent on sustained attention tasks [116,117,118,119,120]. Power increases were positively correlated with a decline in task performance (e.g., increased reaction time and error rates) as well as with subjective state-fatigue ratings [117]. Beside power changes, weakened fronto-parietal coupling in the alpha band, as well as increases in characteristic path length in the alpha and theta band, pointing to a less efficient information transfer, have been shown with cognitive fatigability [121,122,123]. Likewise, the examination of oscillatory patterns in people with MS showed an impaired connectivity balance in a parieto–occipito–temporal network in the alpha and beta band in patients with subjective trait-fatigue measured by total MFIS score [115]. The level of subjective trait-fatigue in patients has further been shown to correlate positively with increased resting state functional connectivity between frontal regions in the theta and beta band as well as with an anterior–posterior increase in beta band connectivity [114]. Thus, progressive power increases and connectivity distortions in low frequency bands have been interpreted as possible indices of cognitive fatigability. However, alpha and theta activity have also been assigned a positive functional role in maintaining an alert state. Theta power and theta band phase synchronization between medial and lateral prefrontal areas have been shown to increase following errors or negative feedback on sustained attention tasks [124,125]. This suggests a central role of theta band activity in cognitive control during prolonged cognitive testing. Furthermore, correlations between frontal activity and posterior alpha power may point to a modulatory role of theta-driven frontal activity on alpha oscillations in sensorimotor regions thereby controlling activity of task-relevant and -irrelevant brain regions [126,127]. Thus, levels of alpha and theta activity might rather be interpreted as an indicator of increased effort to maintain an alert state [116,126]. Specifically, disturbed coupling in these frequency bands might be a crucial factor leading to fatigability-related performance declines.

Assuming that the reported low frequency oscillatory patterns play a mechanistic role in the pathogenesis of cognitive fatigue, the manipulation of abnormal oscillations by tACS might be a central aspect of effective fatigue treatment. tACS involves the application of rapidly alternating electrical currents to the scalp and is assumed to cause periodic shifts in membrane potential and an entrainment (i.e., temporal phase alignment), of neural activity to the externally applied current [128,129]. Although the direct assessment of neural tACS effects in humans is still complicated by electrical artifacts in concurrent neural recordings, findings on behavioral effects during stimulation and analyses of electrophysiological stimulation aftereffects provide good evidence for the efficacy of tACS to modulate oscillatory activity in a phase- and frequency-dependent manner [130,131,132]. Lasting power and connectivity changes at the stimulation frequency have been interpreted as spike-timing-dependent plasticity effects as a consequence of synchronized activity during stimulation [131,133]. To our knowledge, research on MS-related fatigue has not made use of the neuromodulatory potential of tACS so far. 

Findings on a disturbed connectivity pattern within the fatigue circuit, together with oscillatory changes in low-frequency bands with time-on-task in healthy subjects, motivate the possible application of various tACS montages. As frontal theta activity has been related to monitoring of cognitive processes, tACS applied at low frequencies might be used to increase frontal cognitive control and to counteract performance decline over time. Based on findings of interrelations between theta and alpha activity, tACS applied in the theta range might also improve regulation of sensorimotor alpha power. In a recent tACS study on cognitive fatigue in healthy subjects, Loeffler et al. [98] applied tACS in the gamma range during a vigilance task with the aim to decrease inhibitory alpha power over task-relevant cortical regions via cross-frequency interactions. Results showed that gamma tACS counteracted the reaction time increase with time-on-task, yet, effects on occipital alpha power remained speculative due to missing EEG recordings. In a study by Clayton et al. [99], alpha tACS applied to the parieto-occipital cortex during sustained attention tasks has been shown to have an overall stabilizing effect on performance level with time-on-task. This result might support the notion that increased alpha activity does not merely reflect a decrease in attention state. As synchronized, low-frequency activity seems to play an important role in maintaining cognitive control, bifocal tACS applied in the theta range over frontal cortices or in the alpha range over frontal and parietal areas might be effective in counteracting disturbed coupling typically seen with increasing fatigue levels [115,126]. 

Overall, fatigue relates to a complex brain state involving multiple brain regions within the fatigue circuit. The current literature suggests an important role of alterations in local excitability as well as oscillatory activity and connectivity inside the fatigue network that might result from demyelination and axonal degeneration in MS. Even if speculative, hyperactivity in the fatigue network might be related to cognitive control and an increased attentional effort to maintain an attentional state. However, with time-on-task this overactivation might not be sufficient to compensate for processing inefficiencies and coupling alterations in other parts of the network. This assumption implies a central role of connectivity patterns in fatigue pathogenesis and could explain variability in the efficacy of tDCS to counteract fatigue symptoms. The complex nature of fatigue-related neural mechanisms might be more holistically treated by taking into consideration local excitability deficiencies as well as altered connectivity within the whole fatigue circuit. Therefore, the use of tACS complementary to tDCS might help to decode cognitive processes underlying cognitive fatigue. A combination of brain stimulation and neuroimaging techniques might be best suitable to test the effects of tES protocols on local and global activity changes inside the fatigue circuit. For the development of patient-tailored stimulation protocols, it is essential to further investigate the variability in responsiveness to tES application among people with MS. Different patterns of brain damage and anatomical differences in the tES target region might lead to variable stimulation efficiency. In previous tDCS studies on subjective fatigue in MS, tDCS effect size has not been found to correlate with demographic characteristics of patients like age, disease duration, or disability [107,111]. Positive correlations have been reported for tDCS efficiency with lesion load in the left frontal cortex as well as with baseline fatigue levels [106,107,108]. Interestingly, Ferrucci et al. [111] reported that the subgroup of responders was significantly younger than non-responders. This result might suggest that therapeutic benefits of tES might require residual metabolic activity leaving more space for functional improvements [134].

## 6. Conclusions and Outlook

Fatigue is one of the most common symptoms encountered in people with MS and the main cause of early retirement. Thus, the development of reliable diagnostic instruments is of utmost clinical and social relevance. Recent investigations to complement the subjective nature of fatigue diagnostics by objective fatigue measures (i.e., simple reaction time or P300 ERP), are an important step to an integral diagnostic process and treatment evaluation. While the value of fatigability parameters has previously been critically discussed based on its inconsistent relation with subjective fatigue levels, we emphasize that objective manifestations of fatigue should not substitute subjective fatigue assessment but complement it in fatigue diagnostics. The MS fatigue construct is as complex as its underlying neural causes and should be diagnosed and treated in a holistic manner. Differentiating between individual aspects of the fatigue construct and a clear referencing to the taxonomy in scientific communication will help to provide clarity in further research on MS fatigue. In the absence of a common MS fatigue therapy, neuromodulation by tES provides a promising alternative treatment approach and additionally enables the causal investigation of underlying pathological mechanisms. Since tES methods are economic, easy to apply, and well tolerated, they allow for a large-scale use in clinical practice. Former evidence for improvements in fatigue symptoms by tES application encourages further investigation of effective and patient-tailored stimulation protocols. 

## Figures and Tables

**Figure 1 brainsci-09-00100-f001:**
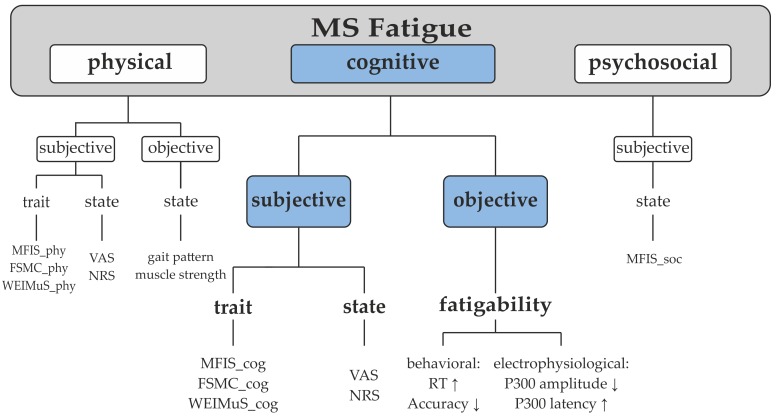
Fatigue classification. MFIS, Modified Fatigue Impact Scale; MS, multiple sclerosis; FSMC, Fatigue Scale for Motoric and Cognitive Functions; WEIMuS, Wuerzburg Fatigue Inventory for Multiple Sclerosis; VAS, Visual Analogue Scale; NRS, Numerical Rating Scale; RT, reaction time; ↑, increase; ↓ decrease.

**Table 1 brainsci-09-00100-t001:** Overview of studies investigating objective cognitive fatigability in people with multiple sclerosis (MS).

Reference	Parameter	Sample Size	EDSS Score	Duration of MS (in Years)	Conceptualization	Fatigability	Correlation with Subjective Fatigue
**Cognitive fatigue over an extended time**							
Andreasen et al., 2010 [34]	Processing speed	60 MS (all RR), 18 HC	PF: 3.0 (1–3.5) ^b^ SF: 2.5 (2–3.5) ^b^ NF: 2 (1.5–−3.5) ^b^	PF: 5.0 (1–14) ^b^ SF: 3.5 (0–16) ^b^ NF: 3.0 (0–9) ^b^	Processing speed across two testing blocks	No: processing speed improved in second testing block	Yes: negative correlation between subjective trait-fatigue and cognitive performance
Bailey et al., 2007 [32]	RT, Accuracy	14 MS (all PP + SP), 17 HC	7.7 (0.4) ^a^	27.2 (8–59) ^c^	Performance in 0-back (attention) and 1-back task (working memory)	Yes: accuracy decreased over time	No: no correlation between subjective state-fatigue and fatigability
Beatty et al., 2003 [35]	Processing speed	17 MS (13 RR, 4 SP), 12 HC	2.9 (2.3) ^a^	14.2 (7.4) ^a^	Performance in cognitive tests (list recall, letter-number sequence, SDMT, PASAT) before and after workday	No: no performance decline from first to second testing block	No: no correlation between subjective state-fatigue and cognitive performance after workday
Bruce et al., 2010 [36]	RT, RT variability	87 MS (70 RR, 17 SP), 24 HC	4.5 (1.6) ^a^	10.9 (7.9) ^a^	Performance across three blocks of CARB	No: shorter RT and smaller variability over time	Yes: positive correlation between subjective trait-fatigue and cognitive performance
Claros-Salinas et al., 2010 [33]	RT	20 MS, 76 HC, 22 stroke	-	8.2 (7.2) ^a^	Performance in three TAP subtests at three different time points of the day	Yes: cognitive performance decreased over time only in MS patients	Not mentioned
DeLuca et al., 2008 [63]	RT	15 MS (12 RR, 3 PP), 15 HC	-	6.4 (4.9) ^a^	Performance across four blocks of modified SDMT	No: faster RT over time	Not mentioned
Fiene et al., 2018 [25]	RT, P300 amplitude and latency	15 MS (14 RR, 1 SP)	3.5 (1.9) ^a^	9.6 (8.6) ^a^	Performance across three blocks of SRT and auditory oddball paradigm	Yes: increasing RT, shorter amplitude and longer latencies of P300 over time	Yes: correlation between subjective state-fatigue and fatigability (negative with P300 amplitude and positive with latency)
Huolman et al., 2011 [59]	Processing speed, RT	15 MS (all RR), 13 HC	1.5 (0.9) ^a^	4.2 (3.6) ^a^	Performance of the last 20 items across four blocks of a modified version of the PVSAT	Yes: group differences increased over time	Not mentioned
Johnson et al., 1997 [37]	Processing speed	15 MS, 15 CFS, 15 MD, 15 HC	1.8 (1.2) ^a^	-	PASAT performance across four testing blocks	No: performance unchanged across blocks	Not mentioned
Sandry et al., 2014 [68]	RT, Accuracy	32 MS (24 RR, 1 PP, 3 SP, 1 PR),24 HC	AI *: 2.4 (2.5) ^a^	11.9 (7.1) ^a^	Task performance (processing speed and working memory task) across four testing blocks	No: RT improved across blocks, no changes in accuracy	No: no correlation between subjective state-fatigue and cognitive performance across blocks
**Cognitive fatigue after challenging mental or physical exertion**							
Claros-Salinas et al., 2013 [47]	RT	32 MS (20 RR, 2 PP, 10 SP), 20 HC	3.6 (1.6) ^a^	7.7 (5.4) ^a^	Performance in TAP subtests before and after physical and cognitive load for 2.5 hours	Yes: people with MS showed a significant increase in RT after cognitive load	Yes: positive correlation between subjective trait as well as state-fatigue and fatigability
Jennekens- Schinkel et al., 1988 [62]	RT	39 MS (20 RR, 19 PP + SP), 25 HC	3.5 (0–7) ^c^	12.0 (1–48) ^c^	Performance in SRT before and after neuropsychological assessment for four hours	No: no group differences in task-related performance decline	Not mentioned
Krupp & Elkins, 2000 [38]	Neuropsychological test battery	45 MS (24 RR, 8 PP, 13 SP), 14 HC	3.8 (1.7) ^a^	-	Performance in neuropsychological test battery before and after cognitive demanding task	Yes: performance of people with MS worsened after cognitive task	Not mentioned
Neumann et al., 2014 [39]	RT	30 MS (23 RR, 1 PP, 6 SP), 15 HC	F: 3.8 (1.2) ^a^ NF: 3.7 (0.6) ^a^	F: 9.9 (6.7) ^a^ NF: 13.6 (6.8) ^a^	Performance in TAP alertness test before and after cognitive load and after a one hour resting time	Yes: increased RT in MS group after cognitive load; after rest RT returned to baseline in most patients	Yes: positive correlation between subjective trait-fatigue and cognitive performance
Paul et al., 1998 [64]	Accuracy, memory performance	39 MS, 19 HC	AI: 4.1 (2.5) ^a^	12.2 (4.8) ^a^	Performance in Word List Learning Task and vigilance test before and after a cognitive work battery that lasted 30 min	No: neither patients nor controls showed changes in cognitive performance after 30 min task	Not mentioned
Spiteri et al., 2017 [69]	RT	40 MS (25 RR, 2 PP, 13 SP), 22 HC	3.5 (1.5) ^a^	14.1 (8.8) ^a^	Performance in alertness test before and after a cognitive demanding task (n-back)	Yes: patients responded slower and with greater variability after n-back task	No: no correlation between subjective trait as well as state-fatigue and cognitive performance
**Cognitive fatigue during sustained mental effort**							
Berard et al., 2018 [70]	Processing speed	32 MS (all RR), 32 HC	1.8 (1.2) ^a^	4.4 (3.1) ^a^	Performance in first third versus last third of PASAT	Yes: poorer performance in last third of PASAT	No: no correlation between subjective trait-fatigue and fatigability
Bryant et al., 2004 [41]	Processing speed	56 MS, 39 HC	-	SG1: 5.8 (1.6) ^a^ SG2: 10.6 (1.8) ^a^	Performance in first versus second half of each of four PASAT testing blocks	Yes: percent dyads declined earlier in time in MS subgroup	No: no correlation between subjective trait-fatigue and cognitive performance
Cehelyk et al., 2019 [60]	RT	21 MS (19 RR, 2 SP)	3.5 (1.6) ^a^	13.3 (8.7) ^a^	Performance in first versus fourth quarter of Blocked Cyclic Naming Task	Yes: RT increased from first to fourth quarter	Yes: positive correlation between subjective state-fatigue and fatigability
Chinnadurai et al., 2016 [42]	Processing speed, P300	50 MS (36 RR, 2 PP, 12 SP), 50 HC	4.6 (1.9) ^a^	6.0 (7.4) ^a^	Performance in 60 and 180 sec version of Stroop Task, SDMT, serial addition task and ratio between first and last 50 items in P300 oddball paradigm	Yes: performance decline and increasing P300 latencies in last 50 items only in people with MS	Not mentioned
Crivelli et al., 2012 [71]	RT	27 MS (all RR), 27 HC	1.03 (0.8) ^a^	0.7 (0.7) ^a^	Performance in third compared to first block of three attentional network tests (alertness, orienting, executive control)	No: performance improved over time	Not mentioned
DeLuca et al., 2008 [63]	RT, Accuracy	15 MS (12 RR, 3 PP), 15 HC	-	6.4 (4.9) ^a^	Performance in second compared to first half in each of four blocks of modified SDMT	No: both groups responded faster in second half of each block	Not mentioned
Gossmann et al., 2014 [43]	Accuracy	31 MS (all RR), 10 HC	3.6 (2.1) ^a^	10.4 (9.2) ^a^	Omissions in second half compared to first half of a 30 min auditory vigilance test	Yes: only in MS group performance declined significantly during the task	Yes: positive correlation between subjective state-fatigue and fatigability
Hanken et al., 2016 [48]	RT	46 MS (18 RR, 28 PP + SP)	LF: 3.7 (1.8) ^a^ HF: 4.7 (1.1) ^a^	LF: 13.5 (8.8) ^a^ HF: 10.9 (7.8) ^a^	Performance in first 5 min compared to last 5 min of a 20 min visual vigilance task	Yes: RT increased with time-on-task	Not mentioned
Kluckow et al., 2016 [51]	Processing speed	36 MS (all RR), 36 HC	1.9 (1.2) ^a^	2.8 (6.6) ^a^	Performance in PASAT during the last 20 items compared to first 20 items and performance change in TVA from first to fourth block	Yes: processing speed of MS group declined in second half of TVA (especially in high-fatigue patients)	Not mentioned
Kos et al., 2004 [44]	Processing speed	50 MS, 21 HC	6.4 (1.2) ^a^	-	Performance in the first ten items compared to the last ten items in PASAT	Yes: 21.1% performance decline in MS group	No: no correlation between subjective trait-fatigue and fatigability
Kujala et al., 1995 [45]	RT, Accuracy	45 MS (22 RR, 17 PP, 6 SP), 35 HC	CP: 5.0 (1.8) ^a^ CD: 5.5 (1.3) ^a^	CP: 8.7 (5.9) ^a^ CD: 8.7 (6.0) ^a^	Performance in visual vigilance test over 15 min	Yes: slower RT with time-on-task; the cognitively preserved MS group also showed decline in accuracy	Not mentioned
Lehmann et al., 2012 [52]	RT, Accuracy	42 MS (all RR), 11 HC	F: 2.8 (1.4) ^a^ NF: 4.3 (2.7) ^a^	-	Performance decline from first to second half of a 10 min 2-back task	No: no task-related performance changes with time-on-task	Not mentioned
Schwid et al., 2003 [50]	Processing speed	20 MS (10 RR, 2 PP, 8 SP), 21 HC	3.8 (1.5) ^a^	-	Performance in first 20 items compared to last 20 items in PASAT	Yes: performance decline over time only in people with MS	Yes: correlation between subjective trait-fatigue and fatigability
Walker et al., 2012 [53]	Processing speed	70 MS (all RR), 70 HC	1.8 (1.2) ^a^	4.4 (3.1) ^a^	Performance during first compared to second half in PASAT and CTIP	Yes: ability of MS group to meet task demands declined over time	Yes: negative correlation between subjective trait-fatigue and fatigability

* AI (Ambulatory Index): is based on a zero-to-nine-point scale and has been shown to be highly correlated with Expanded Disability Status Scale (EDSS) [72]. (**a**) mean (standard deviation); (**b**) median (range); (**c**) mean (range). Abbreviations: AI, Ambulatory Index; CARB, Computerized Assessment of Response Bias; CD, cognitively deteriorated subgroup; CFS, chronic fatigue syndrome; CP, cognitively preserved subgroup; CTIP, Computerized Test of Information Processing; EDSS, Expanded Disability Status Scale; F, fatigued subgroup; HC, healthy controls; HF, high fatigued subgroup; LF, low fatigued subgroup; MD, major depression; MS, multiple sclerosis; NF, non-fatigued subgroup; PASAT, Paced Auditory Serial Addition Test; PF, primary fatigued subgroup; PP, primary progredient MS form; PR, progressive relapsing MS form; PVSAT, Paced Visual Serial Addition Test; RR, relapsing remitting MS form; RT, reaction time; SDMT, Symbol Digit Modalities Task; SF, secondary fatigued subgroup; SG, subgroup; SP, secondary progredient MS form; SRT, Simple Reaction Time Task; TAP, Test Battery for Attentional Performance; TVA, Theory of Visual Attention.

**Table 2 brainsci-09-00100-t002:** Overview of studies evaluating transcranial electrical stimulation (tES) effects on objective cognitive fatigability.

Reference	Parameter	Sample Size	Stimulation Design	Study Design	Results
**tDCS Studies**					
Borragan et al., 2018 [93]	RT, Accuracy	20 HC	Position: DLPFC Parameters: 1.5 mA for 25 min Average current density: 0.06 mA/cm^2^	Three blocks of PVT; Between first and second block of PVT participants performed cognitive demanding working memory task; From the beginning to the second block participants received anodal or sham tDCS (within-subject design)	Anodal tDCS had no impact on behavioral performance decrements over time; tDCS-related interhemispheric shift in cortical oxygenation after stimulation offset
Fiene et al., 2018 [25]	RT, P300 amplitude and latency	15 MS (14 RR, 1 SP)	Position: DLPFC Parameters: 1.5 mA for circa 30 min Average current density: 0.06 mA/cm^2^	Three blocks of SRT task and auditory oddball paradigm; During second block, participants received anodal or sham tDCS (within-subject design)	Anodal tDCS caused a decrease in RT and an increase in P300 amplitude which persisted after the end of stimulation
Hanken et al., 2016 [48]	RT, Accuracy	Study I: 52 HC Study II: 46 MS (18 RR, 28 PP + SP)	Position: right parietal (Study I + II) or right frontal (Study I) Parameters: 1.5 mA for 20 min Average current density: 0.04 mA/cm^2^	Visual vigilance task for 40 min (Study I) or 20 min (Study II); Anodal or sham tDCS for 20 min (between-subject design)	Anodal tDCS counteracted the time-on-task RT decrements (in people with MS and healthy controls)
McIntire et al., 2014 [94]	RT, Accuracy	30 HC	Position: DLPFC Parameters: 2 mA for 30 min Average current density: 0.199 mA/cm^2^	Five blocks of PVT every two hours after initial baseline assessment; Anodal tDCS with placebo gum or sham tDCS with placebo or caffeine gum after 22 h of wakefulness (between-subject design)	Anodal tDCS prevented vigilance decrements over time and led to better subjective ratings of fatigue, drowsiness and energy; Positive effects lasted at least six hours
McIntire et al., 2017 [95]	RT, Accuracy	50 HC	Position: DLPFC Parameters: 2 mA for 30 min Average current density: 0.199 mA/cm^2^	Five blocks of PVT every two hours after initial baseline assessment; Anodal tDCS with placebo gum or sham tDCS with placebo or caffeine gum early or late in the experiment (after 18 or 22 h of wakefulness) (between-subject design)	Anodal tDCS applied early in the experiment led to improved attentional accuracy and RT lasting for six hours
Nelson et al., 2014 [96]	RT, Accuracy	19 HC	Position: DLPFC Parameters: 1 mA for 10 min Average current density: 0.028 mA/cm^2^	Anodal, cathodal, or sham tDCS early (first 10 min) or late (last 10 min) during a 40 min vigilance task (within-subject design)	Especially early anodal and cathodal tDCS significantly improved task performance
Sarasso et al., 2019 [97]	Accuracy	45 HC	Position: PPC Parameters: 1.5 mA for 15 min Average current density: 0.06 mA/cm^2^	Two blocks of a visual vigilance task; Between blocks participants received right-anodal-left-cathodal, right-cathodal- eft-anodal, or sham tDCS (between-subject design)	Right-cathodal-left-anodal tDCS counteracted the time-on-task decrease in performance accuracy
**tACS Studies**					
Loeffler et al., 2018 [98]	RT, Accuracy	24 HC	40 Hz gamma tACS over visual cortex Parameters: 1 mA for 30 min	tACS was applied during the second block of a vigilance task (the first block taken as a baseline)	tACS significantly decreased the time-on-task related slowdown of RT
Clayton et al., 2019 [99]	RT, Accuracy	178 HC in four studies	10 Hz alpha tACS over posterior cortex Parameters: 2 mA for 11 min	Visual and auditory sustained attention task performance across four blocks; 10 Hz, 50 Hz or sham tACS were applied during second and third block	Alpha tACS exerted a stabilizing effect on accuracy and RT and generally limited the slope of performance deteriorations or improvements over time (specific to visual domain)

Abbreviations: DLPFC, dorsolateral prefrontal cortex; HC, healthy controls; MS, multiple sclerosis; PP, primary progredient MS form; PVT, Psychomotor Vigilance Task; RT, reaction time; SP, secondary progredient MS form; SRT, simple reaction time task; tACS, transcranial alternating current stimulation; tDCS, transcranial direct current stimulation

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
