# Peer review of "Cognitive Fatigue in Multiple Sclerosis: An Objective Approach to Diagnosis and Treatment by Transcranial Electrical Stimulation"

_brainsci, 2019, doi:10.3390/brainsci9050100_

Reviewer 1 Report

You approach an important discapacitating aspect experienced by 75% (or more) of MS patients irrespectively of the stage, or clinical type of MS. While your study defines and contributes substantially to the taxonomy of MS-related fatigue, paves as well the path for future studies utilizing tES as a therapeutic tool for this common syndrome. I would like to encourage you to proceed with controlled clinical trials. There are many questions which remain to be explored: repetitive tES? Can a tES modality be applied to specific degree of severity of fatigue as mesaured by some of the techniques you discussed? These comments are presented for future considerations. 

Author Response

We thank the reviewer for his/her comments and the positive evaluation of our manuscript. We totally agree and hope that the present review can stimulate future research on this topic.

Reviewer 2 Report

This is an interesting written review trying to tie in the different studies on the complex subject of fatigue. As stated in the review the focus is on cognitive fatigue as there are many varying aspects to fatigue including physical and psychosocial as outlined in Figure 1.  

In line 78-80 Further, it is important, to distinguish cognitive fatigue, e.g. strong performance decrements in cognitive demanding tasks over time, from an overall cognitive impairment due to MS-related cortical atrophy [30]. How would this be distinguished? 

In line 224-226 To reliably measure fatigability in MS patients, it is however important to differentiate between MS patients with and without fatigue. Fatigue-related deteriorations can otherwise not be distinguished from MS-related cognitive deficits.This aspect should be considered in future studies on cognitive fatigue. Do you have a proposition to distinguish between these two aspects? Is there a research group already trying to make the distinction that can be readily applied in future? 

It was good that the authors outlined the pitfalls in the methods to measure fatigue in section 111 Objective measurement of cognitive fatigue 

In Table 1 there is no information on the people with MS who were recruited in the studies? Are these people all RR-MS? Are the subjects all within a time frame of their last attack? Are the lesions in the same region for the people with MS? The duration of the disease the same amongst subjects within the study and compared amongst studies? 

Please replace 'MS patients' with 'people with MS' in the text. 

Whilst the 2nd half of the review summarised the current use of tACS and tDCS in healthy control subjects (the one study that did look at people with MS did not compare with healthy controls) I would have appreciated a paragraph discussing the caveats of using tACS and tDCS treatments on people with MS - especially outlining at what stage of the disease the people are at before treatment show benefit either minimal or none. Having this addition to the paper I feel contribute direction to the field for treatment and research. 

Author Response

Reviewer #2:

 We are very thankful to the reviewer for her/his constructive evaluation that helped to improve our manuscript.

 In line 78-80 Further, it is important, to distinguish cognitive fatigue, e.g. strong performance decrements in cognitive demanding tasks over time, from an overall cognitive impairment due to MS-related cortical atrophy [30]. How would this be distinguished?

 The reviewer raises an important point. Cognitive fatigue is defined as a failure to maintain and optimize performance over acute but sustained cognitive effort. Hence, according to this definition it must be operationalized as performance declines over time, rather than as current performance at only one measurement time point. The latter may only reflect the status quo level of overall cognitive impairment due to MS-related atrophies instead of cognitive fatigue. We now rephrased the paragraph in the revised manuscript (p. 2, ll. 74-77).

 In line 224-226 To reliably measure fatigability in MS patients, it is however important to differentiate between MS patients with and without fatigue. Fatigue-related deteriorations can otherwise not be distinguished from MS-related cognitive deficits.This aspect should be considered in future studies on cognitive fatigue. Do you have a proposition to distinguish between these two aspects? Is there a research group already trying to make the distinction that can be readily applied in future?

 We are thankful to the reviewer for pointing on this missing information. Without defining the clinical significance of fatigue symptoms, strong fatigue-related performance declines cannot be distinguished from normal performance decrements over time that can also occur in MS patients without fatigue and healthy controls. Kluger et al. (2013) proposed two common ways for defining the clinical significance of fatigue either by using cut-off values on subjective fatigue questionnaires or by assessing statistically significant differences in fatigability levels between patients and healthy controls. We added this information to the revised version of the manuscript (p. 9-10, ll. 235-246).

 In Table 1 there is no information on the people with MS who were recruited in the studies? Are these people all RR-MS? Are the subjects all within a time frame of their last attack? Are the lesions in the same region for the people with MS? The duration of the disease the same amongst subjects within the study and compared amongst studies?

 As suggested by the reviewer, Table 1 now includes detailed information on the MS form, physical disability status and disease duration of all samples investigated. Only Chinnadurai et al. (2016) reported significantly higher EDSS scores in patients showing fatigability with time-on-task. Jennekens-Schinkel et al. (1988) included measures of lesion load. They found lesion load to be associated with reaction time differences at baseline, but reported no correlation between lesion load and reaction time increase after mental effort. Further, regarding the patients’ last attack, only 14 of the 29 studies report the patients’ time frame since the last relapse. Inclusion criteria were a minimum of four weeks (N=10), six weeks (N=2) or twelve weeks (N=2) since the last relapse. We agree with the reviewer on the importance of considering variability in demographic characteristics of patients when investigating the development of objective fatigue symptoms. We now consider this point at the end of section 3. Objective measurement of cognitive fatigue, (p. 10, ll. 242-246).

 “Moreover, closer investigations on the relation between the level of objective cognitive fatigue and demographic characteristics of patients, e.g. disease duration or disability status, can help to understand the implications of this symptom for patients’ daily functioning during the course of the disease.”

 Please replace 'MS patients' with 'people with MS' in the text.

 We now replaced the term 'MS patients' by 'people with MS' in the revised manuscript.

 Whilst the 2nd half of the review summarised the current use of tACS and tDCS in healthy control subjects (the one study that did look at people with MS did not compare with healthy controls) I would have appreciated a paragraph discussing the caveats of using tACS and tDCS treatments on people with MS - especially outlining at what stage of the disease the people are at before treatment show benefit either minimal or none. Having this addition to the paper I feel contribute direction to the field for treatment and research.

 We thank the reviewer for raising this important aspect. In the current literature, it is still unclear why some patients respond to tES treatment while other do not. We agree that a deeper understanding of the tES efficiency in different patient subgroups is of utmost importance for future research. We now included a section discussing variability in tES responsiveness in previous tDCS studies on subjective fatigue in MS and its dependence on patients’ demographic characteristics (p.13, ll. 426-436).
